# Using machine learning to assess the extent of busy ambulances and its impact on ambulance response times: A retrospective observational study

Lars Eide Næss[1,2,3]*, Andreas Jørstad Krüger[1,2,3], Oddvar Uleberg[2,4], Helge Haugland[1,2,3], Jostein Dale[2], Jon-Ola Wattø[2], Sara Marie Nilsen[5,6], Andreas Asheim[5,7]

1 Department of Research and Development, The Norwegian Air Ambulance Foundation, Oslo, Norway, 2 Department of Emergency Medicine and Pre-Hospital Services, St. Olav's University Hospital, Trondheim, Norway, 3 Department of Circulation and Medical Imaging, Norwegian University of Science and Technology, Trondheim, Norway, 4 Division of Emergencies and Critical Care, Department of Research and Development, Oslo University Hospital, Oslo, Norway, 5 Center for Health Care Improvement, St. Olav's University Hospital, Trondheim, Norway, 6 Department of Clinical and Molecular Medicine, Norwegian University of Science and Technology, Trondheim, Norway, 7 Department of Mathematical Sciences, Norwegian University of Science and Technology, Trondheim, Norway

* lars.eide.ness@stolav.no

## Abstract

### Background

Ambulance response times are considered important. Busy ambulances are common, but little is known about their effect on response times.

### Objective

To assess the extent of busy ambulances in Central Norway and their impact on ambulance response times.

### Design

This was a retrospective observational study. We used machine learning on data from nearby incidents to assess the probability of up to five different ambulances being candidates to respond to a medical emergency incident. For each incident, the probability of a busy ambulance was estimated by summing the probabilities of candidate ambulances being busy at the time of the incident. The difference in response time that may be attributable to busy ambulances was estimated by comparing groups of nearby incidents with different estimated busy probabilities.

### Setting

Medical emergency incidents with ambulance response in Central Norway from 2013 to 2022.

**Data Availability Statement:** Data cannot be shared publicly because the use of location and time stamps combined with information on

ambulance activity and incident type could be used to identify personal information, restricted by legal and ethical regulations according to The European General Data Protection Regulation (GDPR), The Norwegian Personal Data Act, The Norwegian Health Register Act and The Norwegian Health Research Act. To get access to data, an ethical approval from the regional ethics committee, REK midt (https://rekportalen.no/#hjem/home) must be given (rek-midt@mh.ntnu.no). Given reasonable request and ethical permission, access to data could be granted by the regional health authority, Central Norway Regional Health Authority, P.o. box 464, N-7501 Stjørdal, Norway (postmottak@helse-midt.no). We have chosen to publish code for our analysis on GitHub, so that our method could be validated using the same data, or similar data from other emergency medical communication centers and ambulance services. A DOI for the GitHub repository including the code has been created at: https://doi.org/10.5281/zenodo.10409831 The files could also be accessed directly at: https://github.com/andreasasheim/Ambulance_response.

**Funding:** LEN was supported by The Norwegian Air Ambulance Foundation (SNLA, https://norskluftambulanse.no/) as part of a PhD project (grant number not applicable). The funder had no role in the design of the study, data collection, analysis, decision to publish, or preparation of the manuscript. AA and SMN were supported by The Norwegian Research Council (forskningsradet.no) grant number 295989. The funder had no role in the design of the study, data collection, analysis, decision to publish, or preparation of the manuscript.

**Competing interests:** The authors have declared that no competing interests exist.

**Abbreviations:** CI, confidence interval; EMCC, emergency medical communication centre; EMS, emergency medical services.

## Main outcome measures

Prevalence of busy ambulances and differences in response times associated with busy ambulances.

## Results

The estimated probability of busy ambulances for all 216,787 acute incidents with ambulance response was 26.7% (95% confidence interval (CI) 26.6 to 26.9). Comparing nearby incidents, each 10-percentage point increase in the probability of a busy ambulance was associated with a delay of 0.60 minutes (95% CI 0.58 to 0.62). For incidents in rural and urban areas, the probability of a busy ambulance was 21.6% (95% CI 21.5 to 21.8) and 35.0% (95% CI 34.8 to 35.2), respectively. The delay associated with a 10-percentage point increase in busy probability was 0.81 minutes (95% CI 0.78 to 0.84) and 0.30 minutes (95% CI 0.28 to 0.32), respectively.

## Conclusion

Ambulances were often busy, which was associated with delayed ambulance response times. In rural areas, the probability of busy ambulances was lower, although the potentially longer delays when ambulances were busy made these areas more vulnerable.

## Introduction

Demand for ambulance services has increased worldwide over the last decades [1–4]. The increasing workload has placed pressure on these services [5], resulting in prolonged response times [6]. In Norway, the number of acute ambulance missions increased from 181,427 in 2011 to 335,316 in 2022 [7].

Ambulance response times may impact morbidity and mortality, especially among patients with time-critical conditions, such as 'the first-hour quintet': cardiac arrest, chest pain, stroke, severe respiratory failure and severe trauma [8]. Based on a study among patients with cardiac arrest [9], a response time of eight minutes or less has been widely accepted as the gold standard [10, 11]. Response time, as a performance measure, is considered an important quality indicator for emergency medical services (EMS) [12]. Several ambulance organisations have specified local or national regulations or guidelines for response time limits [13]. Norway has a diverse geography and demography with large sparsely populated areas and long travel distances [14], making the eight-minute goal a major challenge. There is currently no regulation on ambulance response time in Norway, but according to official guidelines, the service aims to reach 90% of urban and rural patients within 12 and 25 minutes respectively [15].

In a recent paper, ambulances in an urban–rural region in Norway were found to be busy close to 50% of the time during regular weekday working hours, and the authors concluded that resources should be shifted from low-demand to high-demand areas when busy fraction and system load increased [16]. In practice, ambulance services operate in a dynamic environment, where the system adapts by shifting resources if needed. Therefore, it may not be apparent how busy ambulances, as they appear in logistics data, affect response time. The aim of this study was to investigate the extent of busy ambulances in Central Norway and how response times were affected. In particular, we aimed to study the potential trade-off between flexibility and workload in urban areas compared to rural areas.

## Materials and methods

### Setting

Central Norway has a population of 747,000 [17], with a geography comprising urban areas around the main cities and large rural areas and sparsely populated mountainous and coastal regions. Specialised health services in the region are provided by nine public hospitals (Trondheim, Orkanger, Røros, Levanger, Namsos, Ålesund, Molde, Kristiansund and Volda), all owned by the regional health authority (Central Norway Regional Health Authority). St. Olav's University Hospital in Trondheim is the regional tertiary referral and trauma centre.

The regional road ambulance services have a total of 65 ambulance stations (Fig 1). EMS in the region are also provided by general practitioners, ambulance boats, air ambulance helicopters, and a Search and Rescue helicopter service. There are three regional emergency medical communication centres (EMCCs) in the cities of Namsos, Trondheim, and Ålesund. The EMCCs function as the first point of contact for the public in need of EMS while also coordinating different levels of regional and local EMS responses, including ambulance dispatch. If there is need for an ambulance, the EMCC operator sends the most suitable available resource to the incident, often a road ambulance from the closest ambulance station.

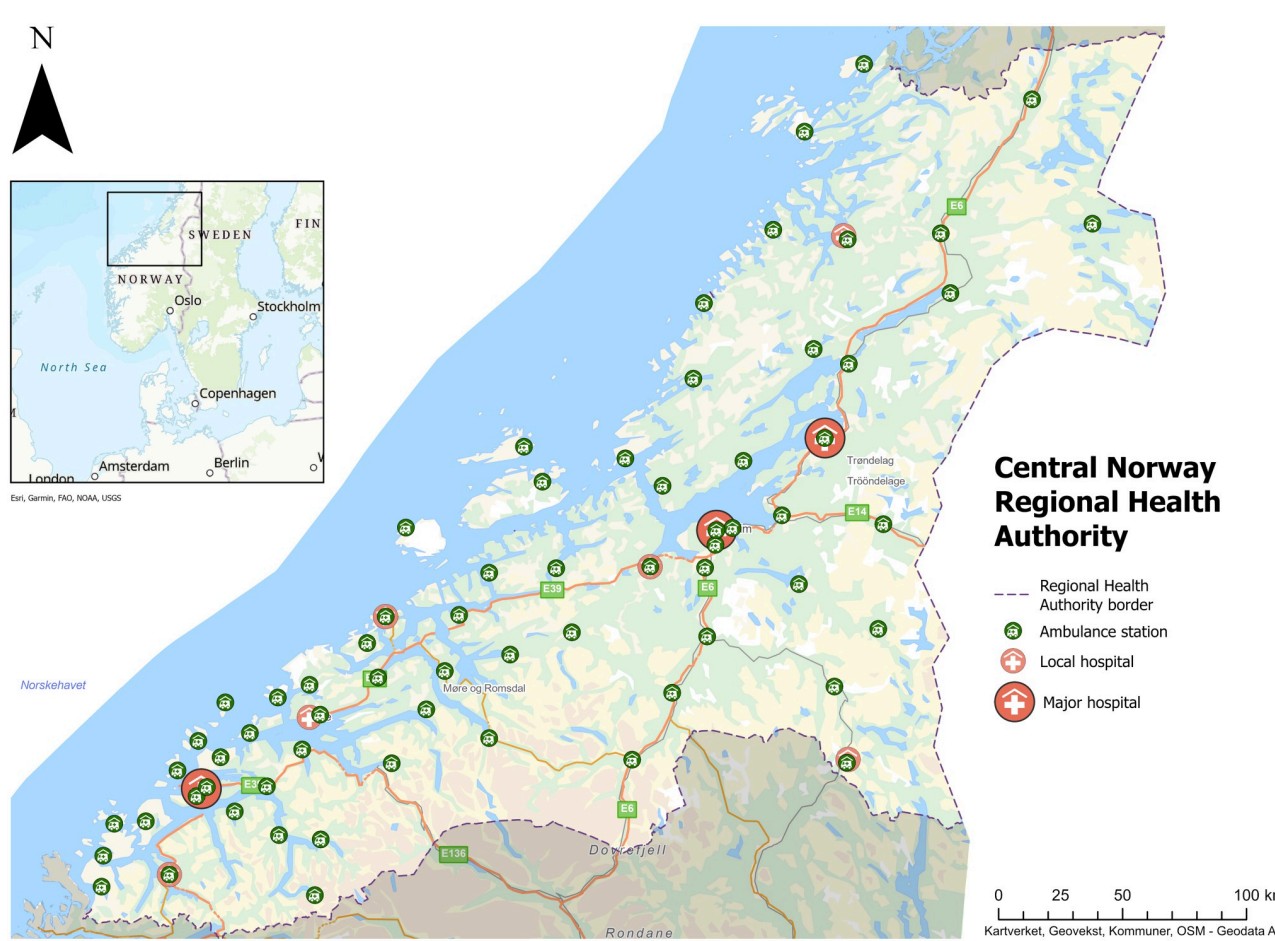

**Fig 1. Ambulance stations and hospital locations in the Central Norway region, 2022.** Background image reprinted from https://geodata.no (Esri, Kartverket, Geovekst, Kommuner, OSM, USGS, Garmin, FAO, NOAA) under a CC BY 4.0 license, with permission from Geodata AS, original copyright 2023.

Nevertheless, the services operates in a dynamic setting where the EMCC operator's decision may depend on circumstances. Several ambulances may service the same area, especially in densely populated areas, and ambulances on transport missions may be re-routed to attend medical emergency incidents. When needed, ambulances from farther away may be dispatched. In situations where no ambulances are available, the EMCC operator could dispatch alternative resources, such as first responders, general practitioners on call or the air ambulance service [18].

## Data and study cohort

In this study, we used information on medical emergency incidents and ambulance missions from the region's EMCC information system. Medical emergency incidents are classified as ordinary, urgent or acute by the EMCC operator according to a national triage scale (Norwegian Index for Medical Emergency Assistance) [19]. We focused on acute incidents, as they represent potentially time-critical medical situations. The data included the type of incident, geographical coordinates, information on dispatched resources, and an identifier for dispatched ambulance units. The data source also included time stamps (dd.mm.yyyy/hh:mm) for time of the call, dispatch, scene arrival and end of mission. Incident coordinates were either gathered automatically through Advanced Mobile Location, registered by the operator through lookup in databases on street address or geographical points of interest, or set manually by the EMCC operator. The time of an incident was defined as the first among possibly several separate calls to the EMCC. Ambulance response time was defined as the difference between the time of the incident and the arrival of the first responding ambulance on the scene. Based on the geographic coordinates, we determined the neighbourhood areas for all the incidents. Neighbourhoods are the least aggregated geographical unit in Norway, defined as geographically coherent areas with up to 300–400 inhabitants in rural areas and 500–600 inhabitants in urban areas [20]. A neighbourhood is defined as urban if it is part of a densely populated area with more than 10,000 inhabitants [21].

From the EMCC incident database, we acquired information on all 278,693 acute medical emergency incidents that had an ambulance response from 1 January 2013 to 31 December 2022. We included only primary missions, i.e., missions that involved transport from the scene to a healthcare institution. Secondary missions, such as transfers between healthcare institutions, were therefore excluded. We also excluded incidents with missing coordinates and probable errors in time stamps, such as negative or longer than six-hour response time. An inclusion flow chart is shown in Fig 2.

To assess the busy status of the ambulance fleet, we used information on the missions registered in the EMCC information system, providing information on all 896,659 ambulance missions during the period. This dataset included both primary and secondary missions at all levels of urgency, ambulance identifiers, and mission start and end times.

The data underlying this study could not be shared publicly due to ethical restrictions on data that might compromise the privacy of individuals. The data may be made available by Central Norway Regional Health Authority upon reasonable request. The regional ethics committee (REK midt) assessed the project, reference number 283508, concluding that the study could be carried out and published without formal approval from REK, in accordance with §2 and §4 of the Norwegian Health Research Act.

The data were obtained from the regional health authority data warehouse server on 10 January 2023. The authors did not have access to any data that could directly identify any participant.

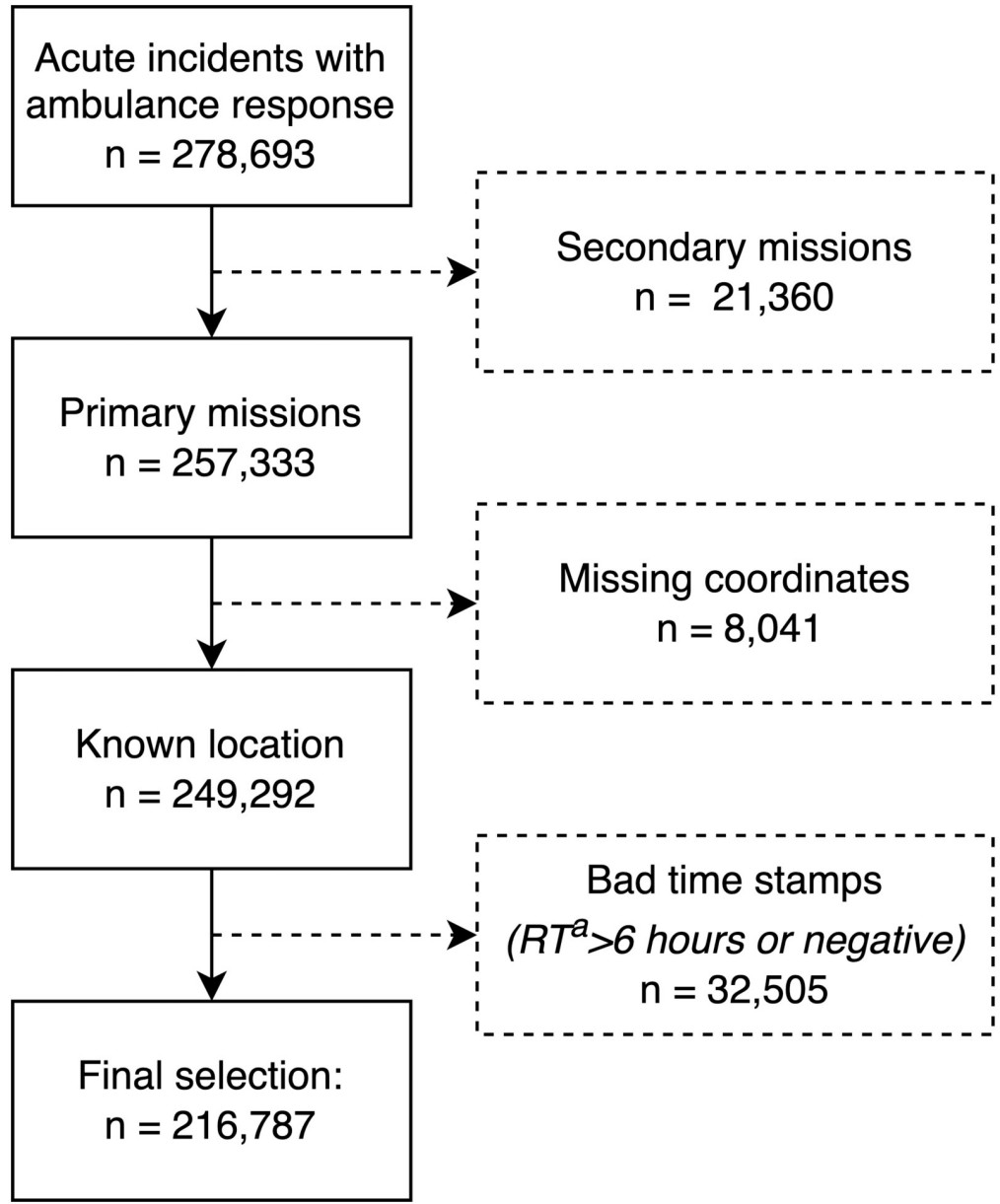

**Fig 2. Data inclusion flow chart.** [a] RT = response time.

## Measures

**Probability of busy ambulances.**   Since the data had no direct information on the alternative choices the EMCC operator could make, we constructed an empirical measure of busy ambulances. The aim was to make a plausible assessment of which ambulances could be candidates to respond to an incident. A straightforward choice would be to pick the last responder to the same area. However, in most cases, more than one ambulance operates in the same area, often with different operating hours. We therefore used machine learning to pinpoint more precisely candidates for responding. Per incident, we used a multinomial regression model [22] for the probability of response from up to five ambulances on other days, modelling the

probability of response from each ambulance, $A_i$, $i = 1,\ldots,5$, relative to an arbitrarily chosen reference ambulance $A_K$,

$$\log\left(\frac{P(A_i|t,d)}{P(A_K|t,d)}\right) = \beta_0^i + \sum_{t_j \epsilon T} \beta_{t_j}^i t_j + \sum_{d_k \epsilon D} \beta_{d_k}^i d_k, \quad i = \{1,\ldots,5\}/K. \tag{1}$$

This model was chosen, because it allows for time of the day and the day of the week, as coded as dummies $t_j, d_k$, to be used as covariates, accounting for possible differences in ambulance operating hours. The coefficients in the model were estimated using feed-forward neural networks based on the 5,000 closest incidents within a radius of 15 km, excluding incidents on the same day. This was a pragmatic choice to ensure that only incidents that occurred close by were considered in both sparsely and densely populated areas.

From the coefficients of the model, the probability of each potential ambulance being candidate to respond to the incident was calculated, and the probability of a busy ambulance was computed as follows:

$$P(busy\ ambulance|t,d) = \sum_{Ambulance\ i} P(A_i|t,d) \cdot 1_{Ambulance\ i\ busy} = \sum_{Busy\ ambulance\ i} P(A_i|t,d). \tag{2}$$

Hence, we estimated the probability of a busy ambulance by summing the predicted probabilities of ambulances being candidates for ambulances that were busy at the time of the incident. As an example, an incident has three potential responding units, A, B and C, with probability of being candidates of 0.6, 0.35 and 0.05, respectively. If ambulances A and C were occupied and ambulance B was available, the probability of a busy ambulance would be 0.6 + 0.05 = 0.65. The result of the prediction also provided the number of candidate ambulances for each incident, defined as the number of resources that had a higher than 10% probability of being candidate, in this example, 2 ambulances (A and B).

**Outcomes.** The outcomes of this study were the prevalence of busy ambulances and the difference in ambulance response times associated with busy ambulances.

## Statistical analysis

As response times are highly heterogeneous over time and space, we designed the analysis around incidents that occurred within restricted areas at similar times but with differing probabilities of busy ambulances. Thus, the analysis emulated a target trial in which incidents in the same areas were randomly exposed to busy ambulances [23]. To assess the effects of busy ambulances on response times, we compared differences between incidents that occurred in the same neighbourhood in the same calendar year but with varying probabilities of busy ambulances, ensuring that geographical variability and changes in infrastructure over time would not substantially influence estimates or negatively affect precision. This was our main approach to control the impact of unmeasured confounding factors. The assumption, which we investigated in additional analyses, was that the busy status of candidate ambulances should not be systematically associated with indications for a quick response. To improve precision, we also adjusted for the time variables month, day of the week, and hour of the day.

Estimates of effects on response time were presented as associations with a 10-percentage point increase in the probability of busy ambulances. We also computed an estimate of mean additional response time per incident as the mean difference between actual response time and predicted response time conditional on no busy ambulance.

Overall estimates and estimates for rural and urban neighbourhoods were computed separately. Estimates for incidents with 1, 2 and >2 candidate ambulances, i.e., the number of ambulances with a more than 10% probability of responding, were also included. All

associations were estimated using fixed effects ordinary linear regression from the *fixest* package for R [24].

Precision was evaluated with 95% confidence intervals (CIs) estimated using Bayesian bootstrapping [25]. The analyses were performed using R (R Core Team, version 4.0.5, 2021) and RStudio (RStudio Team, version 1.4.1106, 2021). We applied multinomial logistic regression with the neural network package nnet [26]. Codes for running the analyses are available on GitHub [27].

## Additional analyses

To assess temporal variation in ambulance availability and response time delay, separate results were computed for each day of the week and hour of the day. We also conducted a balance test to compute the association between the indicator of busy ambulances and the six most common types of incidents (acute illness, traffic accident, other accident, incidents involving psychiatric illness or intoxication, fire, and transport to hospital). A significant association would indicate that incidents with and without busy ambulances had different characteristics that could explain differences in response time.

Changes in demand and mode of operation for EMS during COVID-19 has been documented [28, 29]. To address potential bias related to COVID-19, we performed a separate analysis excluding the affected years 2020 and 2021.

## Results

A total of 216,787 primary acute incidents were eligible for analysis, of which 134,116 were in rural and 82,671 in urban neighbourhoods. Basic descriptions of these incidents are presented in Table 1.

**Table 1. Descriptive statistics.**

|  | All incidents (n = 216,787) | Rural incidents (n = 134,116) | Urban incidents (n = 82,671) |
|---|---|---|---|
| Time of year |  |  |  |
| Jan-Mar | 51,052 (23.5%) | 31,372 (23.4%) | 19,680 (23.8%) |
| Apr-Jun | 53,557 (24.7%) | 33,291 (24.8%) | 20,266 (24.5%) |
| Jul-Sep | 54,144 (25.0%) | 33,868 (25.3%) | 20,276 (24.5%) |
| Oct-Dec | 58,034 (26.8%) | 35,585 (26.5%) | 22,449 (27.2%) |
| Day of the week |  |  |  |
| Weekday | 132,956 (61.3%) | 81,479 (60.8%) | 51,477 (62.3%) |
| Weekend | 83,831 (38.7%) | 52,637 (39.2%) | 31,194 (37.7%) |
| Time of day |  |  |  |
| 0000–0800 | 43,349 (20.0%) | 26,757 (20.0%) | 16,592 (20.1%) |
| 0800–1600 | 89,642 (41.4%) | 55,381 (41.3%) | 34,261 (41.4%) |
| 1600–2400 | 83,796 (38.7%) | 51,978 (38.8%) | 31,818 (38.5%) |
| Response time (minutes) |  |  |  |
| Mean (SD) | 15.8 (12.6) | 18.3 (14.0) | 11.6 (8.5) |
| Median (IQR) | 12.2 (10.4) | 14.8 (13.2) | 10.0 (5.6) |
| 90th percentile | 29.1 | 33.3 | 17.7 |
| Number of candidate ambulances |  |  |  |
| Mean (SD) | 2.42 (0.934) | 2.15 (0.842) | 2.85 (0.911) |
| Median (IQR) | 2 (1) | 2 (1) | 3 (1) |
| Incidents with 1 candidate ambulance | 29,168 (13.5%) | 26,513 (19.8%) | 2,655 (3.2%) |

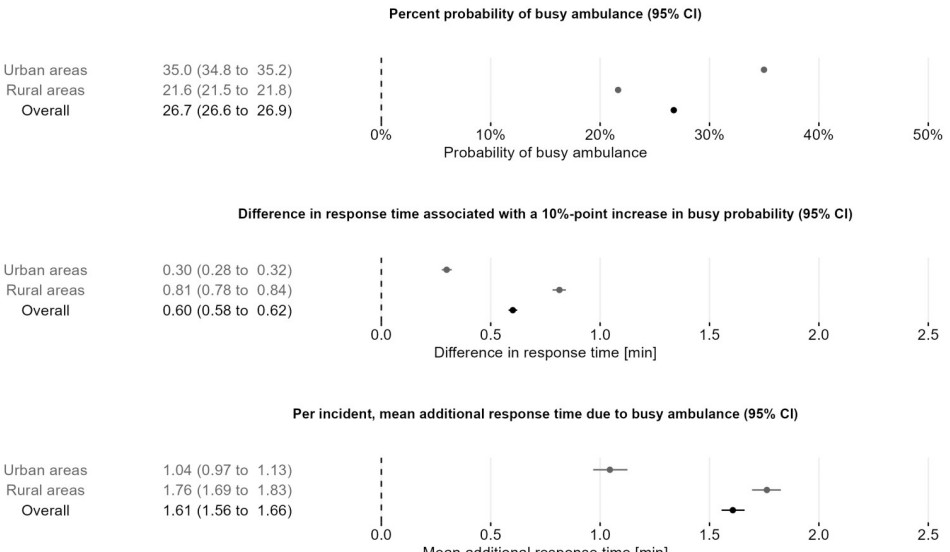

**Fig 3. Probability of busy ambulances, differences in response time with a 10-percentage point increase in busy probability and per incident mean additional response time due to busy ambulances.** Differences in response time were computed within neighbourhood and year and adjusted for hour of the day, day of the week, and month.

The overall estimated probability of a busy ambulance across the whole period and region was 26.7% (95% CI 26.6 to 26.9). A 10 percentage-point increase in the probability of a busy ambulance was associated with an increase in the ambulance response time of 0.60 minutes (95% CI 0.58 to 0.62) when making within neighbourhood-year comparisons. The mean additional response time due to busy ambulances per incident was estimated to be 1.61 minutes (95% CI 1.56 to 1.66) (Fig 3).

For rural incidents, the overall estimated probability of a busy ambulance was 21.6% (95% CI 21.5 to 21.8). A 10-percentage point increase in the probability of a busy ambulance was associated with a 0.81-minute (95% CI 0.78 to 0.84) response time increase, and the per incident mean additional response time due to busy ambulances was 1.76 minutes (95% CI 1.69 to 1.83). The overall estimated probability of a busy ambulance in urban incidents was 35.0% (95% CI 34.8 to 35.2). For these incidents, a 10-percentage point increase in the probability of a busy ambulance was associated with a 0.30-minute (95% CI 0.28 to 0.32) increase in response time. In urban areas, the per incident mean additional response time due to busy ambulances was 1.04 minutes (95% CI 0.97 to 1.13) (Fig 3).

Incidents with more than two candidate ambulances had estimated probabilities of a busy ambulance of 32.6% (95% CI 32.4 to 32.8), whereas incidents with only one candidate ambulance had 17.6% (95% CI 17.4 to 18.0). A 10-percentage point increase in the probability of a busy ambulance was associated with delays of 0.42 (95% CI 0.40 to 0.45) and 1.20 (95% CI 1.12 to 1.27) minutes, respectively (S1 Fig).

## Additional analyses

We found a difference in the overall probability of a busy ambulance from 21% (95% CI 18 to 24) on Mondays to 28% (95% CI 25 to 31) on Sundays (S2 Fig). The delay associated with busy ambulances was longer on Mondays, 0.65 minutes (95% CI 0.58 to 0.72) per 10-percentage point increase in the probability of busy ambulances compared to 0.59 minutes (95%

CI 0.53 to 0.66) on Sundays. Throughout the day, the overall probability of a busy ambulance varied from 12% (95% CI 9 to 16) between 7:00 and 8:00 to 33% (95% CI 29 to 36) between 14:00 and 15:00 (S3 Fig). The delay associated with busy ambulances was longer between 14:00 and 15:00, 0.71 minutes (95% CI 0.58 to 0.84) per 10-percentage point increase in the probability of busy ambulances compared to 0.49 (95% CI 0.31 to 0.66) minutes between 7:00 to 8:00.

There was no substantial association between busy ambulances and the odds of being in one of the six most common incident types (acute illness. accident, transport to hospital, traffic accident, psychiatry or intoxication, fire), as shown in S1 Table.

Excluding data from 2020 and 2021, involving incidents potentially affected by COVID-19, did not substantially change the results (S4 Fig).

## Discussion

In this work, we devised a novel approach to assess the extent of busy ambulances and how it affected ambulance response times in our region. We found that acute medical emergency incidents in urban areas had a higher probability of a busy ambulance than incidents in rural areas. However, response times were more affected by a busy ambulance in rural areas. Incidents in remote areas with one potential responding candidate ambulances were particularly vulnerable. Our results showed that an average of 1.61 minutes of the regional ambulance response time, nearly 10%, could be attributed to busy ambulances. Although we found on average longer response times when ambulances were busy, we could not conclude if the most severe incidents were affected due to lack of patient data.

Challenges in relation to busy ambulances has been documented [16], although the impact of busy ambulances has been less studied. The increasing workload in EMS [1–5] results in more concurrency conflicts [6], and knowledge about the extent and consequences of busy ambulances should be of great importance for EMS administrators. Our results highlight that flexibility in services, i.e. more resources, provides more possibilities to mitigate the negative consequences of busy ambulances. However, this depends on which decisions are taken in situations with pressure on resources. Planning such that ambulances are occupied for similar fractions of the day can lead to rural areas with less flexibility being disadvantaged. A suggested shift of resources from low-demand to high-demand areas in case of increased busy fraction [16] could directly affect the number of candidate ambulances and thus ambulance response times in vulnerable areas.

EMS evolved as an answer to the need for acute health care for certain time-critical conditions (war injuries, cardiac arrest, major trauma) [30, 31], and response time was established as an important service quality indicator [30]. The rationale for swift response times in time-critical situations is well proven, but little is known about the importance of response times for non-time-critical incidents [32]. In contrast to the original purpose of EMS, most of today's medical emergency incidents are not time-critical [6, 33]. In a service with many possible ways of organising a response, prioritising that results in longer response times for less severe incidents is not necessarily a patient safety problem. For many patients, shorter response times will probably not improve survival or other significant outcomes [34]. Depending on the patient's condition and the mission's circumstances, other quality dimensions may be equally or more important. Interventions performed on scene are for instance often more relevant for the patient than the sole effect of reduced response time. Even for high-acuity patients, what truly benefits the patient might not be the swift arrival of an ambulance but that their status and oxygen delivery to tissues are improved. Nevertheless, response time is still one of the most common measures for quality in EMS.

The nature of EMS provides both practical and ethical obstacles for conducting randomised controlled trials, and although access to retrospective observational data on response time is of great use, studying response time data for a wide variety of medical emergency incidents has proven difficult [32, 35]. In the absence of more objective knowledge, subjective interpretation and anecdotal truths have flourished [35], strengthening the common belief that shorter response times are of great importance for all incidents. A purpose of this study was to conduct analyses of response time that could provide a way to utilise real-world data across a large span of incident severities and geographies. Our study approach has potential for studying consequences of delayed ambulance response time in an instrumental variable setting [36], which may contribute to more solid evidence for the future organisation of EMS.

Regardless of how response time affects patient outcomes, response time goals are important for establishing the population's sense of security pertaining to what service to expect when serious illness or trauma occurs. In a region with large sparsely populated areas, timely competence in the prehospital phase of care in remote areas contributes to the equity of access to healthcare for the population. This is supported by a Nordic report on data collection and benchmarking in EMS [30, 37, 38].

Our findings should be interpreted in light of our regional context, and our results are probably not generalisable to other organisational settings. Nevertheless, the methods underlying our findings are applicable to data from any comparable service, and we expect the overall effects to be similar.

## Strengths and limitations

A strength of the analysis was that it relied primarily on well-registered data, objective data, primarily time stamps and coordinates that were collected automatically. With more than 200,000 real-life incidents in ten years, the data provided ample material for precise estimates. Our measure of busy ambulances was calculated from data on the time use of all ambulances. A substantial amount of time could be registered as busy, while the ambulance was effectively available, for example, when returning to base. Therefore, our estimates of the impact of busy ambulances could be an underestimation. This study was based on actual missions, and there was potential for collider bias by requiring an ambulance to have responded. In severe cases where no ambulance was available, other resources would likely have been used. Incidents with missing data were excluded from the study, and bias based on this cannot be ruled out. This study was not designed to assess any effects on patient outcomes.

The ambulance service is a dynamic organisation with continuously shifting resources that adapts to current needs. Therefore, instead of defining fixed resources based on geography alone, we applied machine learning to find expected resources for each response separately. The analyses provided an overall measure of busy ambulances. Imperfections in predicting candidate ambulances might render the approach unsuitable for judging whether a particular incident is subject to busy ambulances. Prediction errors could not be computed, as we had no access to information on actual candidates to respond. When comparing within the same neighbourhood at comparable times, however, it is likely that prediction errors were not substantially associated with incident characteristics. Better sensitivity and specificity of predictions could yield stronger effect estimates, and the presented results may be underestimates.

The study was planned and executed by researchers and clinicians affiliated with the emergency medical services in Central Norway, which may have impacted the assessment of the evidence. However, steps were taken to ensure objectivity, such as specifying the analysis in a published analysis protocol before conducting the analysis [27]. The analysis was based on an approach that mimics a hypothetical randomised experiment [23]. In this approach, we

assumed that ambulance availability was unlikely to differ between incidents of different severities. To the extent that this assumption did not hold, the resulting bias could be in either direction. Without patient data, we were not able to check if case mix could account for any differences in response times associated with busy ambulances, but our balance test showed no relationship between type of incident and probability of busy ambulances. This is an observational study, and other residual sources of bias cannot be ruled out. Associations between patient outcomes, busy ambulances and prolonged response times will be further addressed in future work.

## Conclusions

Ambulances in Central Norway were often busy in the study period, which was associated with delayed ambulance response. In rural areas, the probability of busy ambulances was lower, though the potentially longer delays when ambulances were busy made these areas more vulnerable.

## Supporting information

**S1 Fig. Probability of busy ambulances, differences in response time with a 10-percentage point increase in busy probability and mean additional response time due to busy ambulance per number of candidate ambulances.** Candidate ambulance is defined as an ambulance with more than 10% probability of responding to an incident.
(TIF)

**S2 Fig. Probability of busy ambulances and differences in response time with a 10-percentage point increase in busy probability per day of the week.** Differences in response time were computed within the neighbourhood and year and adjusted for hour of the day, day of the week, and month.
(TIF)

**S3 Fig. Probability of busy ambulances and differences in response time with a 10-percentage point increase in busy probability per hour of the day.** Differences in response time were computed within the neighbourhood and year and adjusted for hour of the day, day of the week, and month.
(TIF)

**S4 Fig. Non-COVID-19 probability of busy ambulances, differences in response time with a 10-percentage point increase in busy probability and per incident mean additional response time due to busy ambulances, excluding the years 2020 and 2021.** Differences in response time were computed within neighbourhood and year and adjusted for hour of the day, day of the week, and month.
(TIF)

**S1 Table. Odds ratio for the incident being one of the six most common incident types associated with a 10% increase in busy probability computed within the neighbourhood and year and adjusted for hour of the day, day of the week, and month.**
(DOCX)

## Acknowledgments

The authors would like to thank Tormod Storsveen Throndsen for his invaluable support in designing Fig 1, and Lars Vesterhus for his knowledgeable inputs on EMCC data.

## Author Contributions

**Conceptualization:** Andreas Asheim.

**Investigation:** Lars Eide Næss.

**Methodology:** Lars Eide Næss, Andreas Asheim.

**Project administration:** Lars Eide Næss, Jostein Dale.

**Resources:** Oddvar Uleberg, Helge Haugland, Jostein Dale, Jon-Ola Wattø.

**Supervision:** Andreas Jørstad Krüger, Andreas Asheim.

**Validation:** Lars Eide Næss.

**Visualization:** Lars Eide Næss, Andreas Asheim.

**Writing – original draft:** Lars Eide Næss, Andreas Asheim.

**Writing – review & editing:** Andreas Jørstad Krüger, Oddvar Uleberg, Helge Haugland, Sara Marie Nilsen.

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
