## [Decision Letter · Decision Letter 0]

3 Oct 2023

PONE-D-23-26575Extent of busy ambulances and the impact on ambulance response times: A cohort studyPLOS ONE

Dear Dr. Næss,

Thank you for submitting your manuscript to PLOS ONE. After careful consideration, we feel that it has merit but does not fully meet PLOS ONE’s publication criteria as it currently stands. Therefore, we invite you to submit a revised version of the manuscript that addresses the points raised during the review process.

We look forward to receiving your revised manuscript.

Kind regards,

Yong-Hong Kuo

Academic Editor

PLOS ONE

Journal Requirements:

3. We note that [Figure 1] in your submission contain [map/satellite] images which may be copyrighted. All PLOS content is published under the Creative Commons Attribution License (CC BY 4.0), which means that the manuscript, images, and Supporting Information files will be freely available online, and any third party is permitted to access, download, copy, distribute, and use these materials in any way, even commercially, with proper attribution. For these reasons, we cannot publish previously copyrighted maps or satellite images created using proprietary data, such as Google software (Google Maps, Street View, and Earth). For more information, see our copyright guidelines: http://journals.plos.org/plosone/s/licenses-and-copyright.

4. We notice that your supplementary [figures/tables] are included in the manuscript file. Please remove them and upload them with the file type 'Supporting Information'. Please ensure that each Supporting Information file has a legend listed in the manuscript after the references list.

Additional Editor Comments:

The manuscript has been reviewed by two referees. Their recommendations are consistent. Overall, they believe that the work has the potential to be published but there are still a number of clarification issues need to be addressed. I recommend a Major Revision for the authors to address their concerns.

Reviewers' comments:

Reviewer's Responses to Questions

**Comments to the Author**

1. Is the manuscript technically sound, and do the data support the conclusions?

Reviewer #1: Yes

Reviewer #2: Partly

2. Has the statistical analysis been performed appropriately and rigorously? 

Reviewer #1: No

Reviewer #2: No

3. Have the authors made all data underlying the findings in their manuscript fully available?

Reviewer #1: No

Reviewer #2: No

4. Is the manuscript presented in an intelligible fashion and written in standard English?

Reviewer #1: Yes

Reviewer #2: Yes

5. Review Comments to the Author

Reviewer #1: I reviewed the article by Lars Eide Næss et al, entitled “Extent of busy ambulances and the impact on ambulance response times: A cohort study” submitted to PLOS ONE (Manuscript ID: PONE-D-23-26575). In this retrospective observational study, the authors mainly assessed the relationship between prevalence of busy ambulances and differences in response times. The authors found that busy ambulances are associated with increased response times, and this effect was particularly evident in rural area. First, the reviewer respects for the Authors' tremendous effort and time spent on this manuscript. However, there are several concerns regarding this manuscript, which are listed below:

1

This reviewer feels that the retrospective observational study is more suitable term for describing this study. Please consider to replace cohort study with retrospective observational study, thought the manuscript.

Methodology

2

The authors stated that "We used machine learning techniques on data from nearby incidents to estimate the weighted probability of up to five different ambulances responding to medical emergencies". This reviewer, as well as many readers are not so familiar with machine learning techniques. Please describe more in details used technique, with relevant references.

2

The ethical statement should include the relevant date, even where the need for approval was waived.

3

Who planned this study, who collected data, and who conducted the statistical analysis? I think if the same researchers are involved in study planning, data collecting, outcome measurement, and statistical analysis, there is a theoretical risk of biased assessment.

4

Describe any efforts to address potential sources of bias. For example, blinded exposure is one attractive method to address this issue.

5

Sample size calculation is missing. Explain how the study size was arrived at. If sample size was not determined a priori, please state so and provide post-hoc sample size estimation to provide the estimation of how was the power of this study.

6

Explain how missing data were addressed.

7

The rationale of the definition of busy ambulance should be described greater in details. Were there any supporting references?

8

Clearly define all potential confounders, and effect modifiers in method section. This study does not consider the any confounders.

9

Please give the characteristic of the data source. How did you do to assure the quality of data? Since this is a retrospective observational study, quality assurance is of vitally important.

Results

10

Table 1

Can the authors provide the patient characteristics (age, year, etc), vital sings, etiology (trauma, cerebral infarction, acute myocardial infarction, etc), and degree of urgency? Such essential data is missing. At this current form, many readers including myself find it difficult to image the characteristics of study subjects. There are too many unmeasured confounders, which hiders the meaningful comparison.

11

Confounders

COVID 19 pandemic can be the significant confounders. The authors should assess the effect of COVID19 pandemic or some other confounders on measured outcomes.

12

The authors state that ambulance response time may impact morbidity and mortality, especially among patients with time-critical conditions. Can author provide the data regarding association among prevalence of busy ambulances, increased response time and patients outcome (e.g. survival rate) in these time sensitive population?

13

Are observed association between prevalence of busy ambulances and differences in response times are clinically meaningful? What is your opinion on this?

Discussion

14

Limitation section needs substantial revision. Please consider the important limitations and do not just list them but consider their relevance and how they might bias the results. Discuss both direction and magnitude of any potential bias.

15

Discuss the generalizability (external validity) and Implications for practice of the study results. Give a cautious overall interpretation of results considering objectives, limitations, multiplicity of analyses, results from similar studies, and other relevant evidence.

16

Please indicate future research direction more in details, immediately after limitation section.

17

Conclusion

The authors concluded that "Using machine learning on observational data to calculate an indicator for busy ambulances showed potential for further research on ambulance response times and patient outcomes" The reviewer thinks this is not objective.

18

The authors should provide the minimal anonymized data set used in this manuscript, according to the journal's policy.

Although the number of criticisms listed above, this reviewer should however state that it is laudable that this work is derived from huge efforts made by the authors, who are working as the frontline healthcare professionals. The reviewer respects the authors’ time and effort spent on this manuscript, and the authors ‘patience and professionalism in dealing with my comments.

Reviewer #2: Thank you for this important subject and this nice paper. However, I believe it could be improved and presented more reproducibly to the audience. Here are some suggestions:

1. You mentioned in the methods, “If there is an acute need for an ambulance, the EMCC operator sends the most suitable available resource to the incident, often a road ambulance from the closest ambulance station. Nevertheless, the services operate in a dynamic setting where the operator’s decision may depend on circumstances. Several ambulances may service the same area, especially in densely populated areas, and ambulances on transport missions may be rerouted to attend acute incidents.” I wonder if it is possible to provide numbers about this time that the operator can take to find the most appropriate unit and its variation and to explain in the method what you did do to account for such confounding factor, specifically here time from receiving an emergency call and finding the most appropriate unit. Also, how much time it takes a unit to move upon receiving the call?

2. You mentioned in lines 247 and 248 that the busy ambulance in a rural area affected the response, which can be understood due to the long distances and extended response time. Hence, I would like to discuss an explanation if this is due to the type of acute medical emergency. Was it affected by some demographic variables like age?

3. Further, you mentioned in the method that you used machine learning technique:

a. This has to be reflected in the title as it is fundamental.

b. Further, I hoped to see in the method and the discussion information about the algorithms you used and why you chose them. This would be important since you used probabilistic methods. Further, to make the study replicable, it is important to mention in the statistics how you handled the missing values and provide a deeper description of your variables and how you evaluated that the model you used is efficient in prediction and how accurate it was. These details are important and give more credibility to the results. These details should also be visible in the abstract.

c. In the limitations, I would advise adding a few sentences about the statistical bias and the limitation of deploying the model you built.

d. This was about prediction; in every model, there would be a true positive, true negative, false positive, and false negative, which was very important to discuss.

e. If you used one model, I don’t think that would be sufficient as another algorithm might perform better, which, unfortunately, you did not mention. So please, I would suggest adding a further discussion about this.

6. PLOS authors have the option to publish the peer review history of their article (what does this mean?). If published, this will include your full peer review and any attached files.

Reviewer #1: No

Reviewer #2: **Yes: **Hassan Farhat

---

## [Author Response · Author response to Decision Letter 0]

14 Nov 2023

Please se attached "Response to reviewers" for a thorough point-by-point response to all comments and suggestions.

---

## [Decision Letter · Decision Letter 1]

10 Dec 2023

Using machine learning to assess the extent of busy ambulances and its impact on ambulance response times: A retrospective observational study

PONE-D-23-26575R1

Dear Dr. Næss,

We’re pleased to inform you that your manuscript has been judged scientifically suitable for publication and will be formally accepted for publication once it meets all outstanding technical requirements.

Kind regards,

Yong-Hong Kuo

Academic Editor

PLOS ONE

Additional Editor Comments (optional):

The authors have successfully addressed the concerns. I recommend Accept.

Reviewers' comments:

Reviewer's Responses to Questions

**Comments to the Author**

1. If the authors have adequately addressed your comments raised in a previous round of review and you feel that this manuscript is now acceptable for publication, you may indicate that here to bypass the “Comments to the Author” section, enter your conflict of interest statement in the “Confidential to Editor” section, and submit your "Accept" recommendation.

Reviewer #1: All comments have been addressed

Reviewer #2: All comments have been addressed

2. Is the manuscript technically sound, and do the data support the conclusions?

Reviewer #1: Yes

Reviewer #2: Yes

3. Has the statistical analysis been performed appropriately and rigorously? 

Reviewer #1: Yes

Reviewer #2: Yes

4. Have the authors made all data underlying the findings in their manuscript fully available?

Reviewer #1: No

Reviewer #2: Yes

5. Is the manuscript presented in an intelligible fashion and written in standard English?

Reviewer #1: Yes

Reviewer #2: Yes

6. Review Comments to the Author

Reviewer #1: Thank you for your time and effort spent on this revision. You addressed my concerns appropriately. There are no remaining comment.

Reviewer #2: Thank you again for this piece of work. Now, with the revisions implemented the manuscript is more coherent and would make a significant contribution to the literature.

7. PLOS authors have the option to publish the peer review history of their article (what does this mean?). If published, this will include your full peer review and any attached files.

Reviewer #1: **Yes: **Yuko Ono, M.D., Ph.D.

Department of Disaster and Emergency Medicine, Graduate School of Medicine, Kobe University

7-5-2 Kusunoki-cho, Chuo-ward, Kobe, 650-0017, Japan

Tel: +81-78-382-6521 Fax: +81-78-341-5254

E-mail: windmill@people.kobe-u.ac.jp

e-Profile: https://researchmap.jp/windmill/?lang=japanese

Reviewer #2: No

---

## [Editor Report · Acceptance letter]

28 Dec 2023

PONE-D-23-26575R1 

PLOS ONE

Dear Dr. Næss, 

I'm pleased to inform you that your manuscript has been deemed suitable for publication in PLOS ONE. Congratulations! Your manuscript is now being handed over to our production team.

Kind regards, 

on behalf of

Dr. Yong-Hong Kuo 

Academic Editor

PLOS ONE